# RNA Sequencing of Medusavirus Suggests Remodeling of the Host Nuclear Environment at an Early Infection Stage

Ruixuan Zhang,[a] Hisashi Endo,[a] Masaharu Takemura,[b] Hiroyuki Ogata[a]

aBioinformatics Center, Institute for Chemical Research, Kyoto University, Gokasho, Japan
bLaboratory of Biology, Institute of Arts and Sciences, Tokyo University of Science, Shinjuku, Tokyo, Japan

**ABSTRACT** Viruses of the phylum *Nucleocytoviricota*, or nucleo-cytoplasmic large DNA viruses (NCLDVs), undergo a cytoplasmic or nucleo-cytoplasmic cycle, the latter of which involves both nuclear and cytoplasmic compartments to proceed viral replication. Medusavirus, a recently isolated NCLDV, has a nucleo-cytoplasmic replication cycle in amoebas during which the host nuclear membrane apparently remains intact, a unique feature among amoeba-infecting NCLDVs. The medusavirus genome lacks most transcription genes but encodes a full set of histone genes. To investigate its infection strategy, we performed a time course RNA sequencing (RNA-seq) experiment. All viral genes were transcribed and classified into five temporal expression clusters. The immediate early genes (cluster 1, 42 genes) were mostly (83%) of unknown functions, frequently (95%) associated with a palindromic promoter-like motif, and often (45%) encoded putative nucleus-localized proteins. These results suggest massive reshaping of the host nuclear environment by viral proteins at an early stage of infection. Genes in other expression clusters (clusters 2 to 5) were assigned to various functional categories. The virally encoded core histone genes were in cluster 3, whereas the viral linker histone H1 gene was in cluster 1, suggesting they have distinct roles during the course of the virus infection. The transcriptional profile of the host *Acanthamoeba castellanii* genes was greatly altered postinfection. Several encystment-related host genes showed increased representation levels at 48 h postinfection, which is consistent with the previously reported amoeba encystment upon medusavirus infection.

**IMPORTANCE** Medusavirus is an amoeba-infecting giant virus that was isolated from a hot spring in Japan. It belongs to the proposed family "*Medusaviridae*" in the phylum *Nucleocytoviricota*. Unlike other amoeba-infecting giant viruses, medusavirus initiates its DNA replication in the host nucleus without disrupting the nuclear membrane. Our RNA sequencing (RNA-seq) analysis of its infection course uncovered ordered viral gene expression profiles. We identified temporal expression clusters of viral genes and associated putative promoter motifs. The subcellular localization prediction showed a clear spatiotemporal correlation between gene expression timing and localization of the encoded proteins. Notably, the immediate early expression cluster was enriched in genes targeting the nucleus, suggesting the priority of remodeling the host intranuclear environment during infection. The transcriptional profile of amoeba genes was greatly altered postinfection.

**KEYWORDS** NCLDV, RNA-seq, giant virus, medusavirus

Giant viruses are characterized by their large viral particles and complex genomes and are found worldwide (1–6). They have been classified within the phylum *Nucleocytoviricota* (also referred to as nucleo-cytoplasmic large DNA viruses [NCLDVs]) (7). Phylogenetic analyses suggested that the diversification of this group of viruses predated the emergence of modern eukaryotic lineages (8, 9), which revived the

Address correspondence to Masaharu Takemura, giantvirus@rs.tus.ac.jp, or Hiroyuki Ogata, ogata@kuicr.kyoto-u.ac.jp.

debate about their evolutionary origin (10, 11) and their relationship to the genesis of the eukaryotic nucleus (12, 13). Genomic analysis revealed a large number of genes (referred to as orphan genes) without detectable homology to any known genes. The abundance of orphan genes or lineage-specific genes has been considered evidence that supports the ongoing *de novo* creation of genes in these viruses (14, 15). In addition to the efforts to isolate and characterize new giant viruses, environmental genomics has revealed their ubiquitous nature, extensive gene transfers with eukaryotes, and complex metabolic capabilities (16–18).

Medusavirus, a giant virus that infects the amoeba *Acanthamoeba castellanii*, was isolated from a hot spring in Japan (2). Recently, a related virus, medusavirus stheno, was isolated from fresh water in Japan (19) and another distantly relate virus, clandestinovirus, was isolated from wastewater in France (20). These three viruses represent the proposed family "*Medusaviridae*" (2), which is distantly related to other giant virus families and forms an independent branch in the tree of the phylum *Nucleocytoviricota*. During the infection cycle of medusavirus, its genome enters the host nucleus to initiate DNA replication, and particle assembly and DNA packaging are carried out in the cytoplasm. Of note, the host nuclear membrane remains intact until near the end of the viral replication cycle, which represents a unique feature of medusavirus among currently characterized amoeba-infecting giant viruses. The viral replication cycles of other amoeba-infecting giant viruses are characterized as either a cytoplasmic replication by establishing cytoplasmic viral factories (e.g., mimiviruses [21], marseilleviruses [22], pithoviruses [5], cedratvirus [23], and orpheovirus [24]) or a nucleo-cytoplasmic replication, like in medusavirus, but with a degradation of the host nucleus (e.g., pandoraviruses [15] and molliviruses [1]). For medusavirus, no visible cytoplasmic virus factory has been observed by transmission electron microscopy (2). Thus, it appears that the host nucleus is transformed into a virus factory, from which mature and immature medusavirus virions emerge. It has also been reported that some of the host amoeba cells display encystment upon medusavirus infection as early as 48 h postinfection (hpi) (2). Medusavirus has a 381-kb genome that encodes 461 putative proteins; 86 (19%) have their closest homologs in *A. castellanii*, whereas 279 (61%) are orphan genes. Compared with other amoeba-infecting giant viruses, medusaviruses have fewer transcriptional and translational genes and have no genes that encode RNA polymerases and aminoacyl-tRNA synthetases, suggesting that medusaviruses are heavily reliant on the host machinery for transcription and translation. In contrast to their paucity in expression-related genes, medusaviruses are unique among known viruses in encoding a complete set of histone domains, namely, the core histones (H2A, H2B, H3, and H4) and the linker histone H1. A virion proteomic study detected proteins encoded by the four core histone genes in medusavirus particles (2). Given these unique features, medusavirus is expected to have a characteristic infection strategy among known amoeba-infecting giant viruses. However, the dynamics of gene expression during the medusavirus infection cycle has not been investigated so far.

Previous RNA sequencing (RNA-seq) studies of giant viruses detected viral genes that were expressed in a coordinated manner during the viral infection. Viral genes that belong to different functional categories tend to show different expression patterns and can be grouped as, for instance, early, intermediate, or late. Different viruses also have different gene expression programs; for example, the transcription order of informational genes (those involved in replication, transcription, translation, and related processes) can differ among viruses. The expression of DNA replication genes (starting from 3 hpi) precedes the expression of transcription-related genes (6 hpi) in mimivirus, whereas this order is reversed in marseillevirus (i.e., transcription-related genes from <1 hpi and DNA replication genes at 1 to 2 hpi) (25, 26). Putative promoter motifs associated with temporal expression groups have been identified in mimiviruses

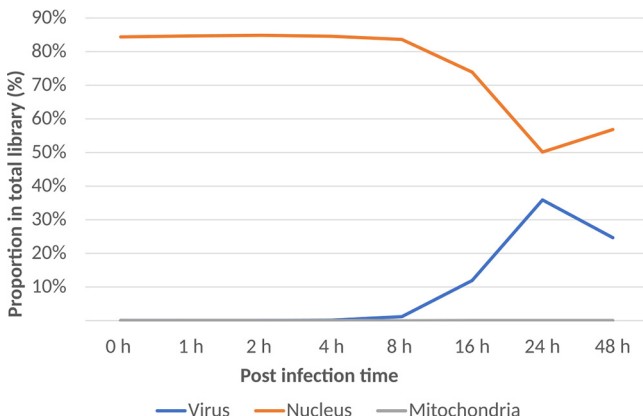

**FIG 1** Proportions of viral and host mRNA reads at different time points during the course of medusavirus infection.

and marseilleviruses (25–28). The expression patterns of host genes during infection of giant viruses have been investigated by RNA-seq and proteomics (1, 26, 28).

We performed a time-series RNA-seq analysis of infected amoeba cells to investigate the transcriptional program and infection strategy of medusavirus. We report expression clusters of medusavirus genes, putative viral promoter motifs, and changes in host gene expression.

## RESULTS

**Transcription profile of medusavirus genes.** The overall composition of the mRNA library during the course of the virus infection is shown in Fig. 1 (see Data set S1 in the supplemental material). Until 8 hpi, viral reads were less than 1% of the total reads, and then they increased and reached a peak at 24 hpi. The proportion of host reads stayed at a high level during the first 8 h and then decreased rapidly and reached a minimum at 24 h, which still accounted for approximately half of the total reads in the library.

All viral genes were gradually expressed and continuously increased up to 16 hpi (Fig. 2A). We identified five clusters of viral gene expression profiles using the *k*-means method (Fig. 2B) and named these clusters as follows: cluster 1 (immediate early) genes showed a gradual increase in expression from 0 hpi; cluster 2 (early) genes showed a gradual increase in expression from 1 hpi; clusters 3 and 4 (intermediate) genes showed a gradual increase in expression from 2 hpi; and cluster 5 (late) genes showed a gradual increase in expression from 4 hpi. The expression patterns of genes in clusters 3 and 4 were only slightly different; genes in cluster 3 showed higher Z-score scaled reads per kilobases of transcript per Million mapped reads (RPKM) values at 8 hpi than those in cluster 4. In the following text, both of these clusters were referred to as "intermediate" genes.

The distribution of genes with annotated functions showed characteristic patterns among the five expression clusters (Fig. 3A to C; Data set S8). Among the 42 genes in cluster 1 (i.e., immediate early), 35 (83%) were unknown genes. Of those annotated in this cluster, there were a linker histone H1 gene and a poly-A polymerase regulatory subunit gene. The proteins encoded by these two genes were not detected in a previous virion proteomic study of medusavirus (2). Cluster 2 included genes that were classified in the "nucleotide metabolism" and "DNA replication, recombination, and repair" categories, including a DNA helicase, a DNA primase, and ribonucleotide reductase large/small subunits. Cluster 3 contained genes in various functional categories, including histone genes (the four core histone genes H2A, H2B, H3, and H4), "DNA replication, recombination, and repair" category (e.g., two of five nuclease genes, a Yqaj viral recombinase gene, and a Holliday junction resolvase gene), "Transcription and RNA

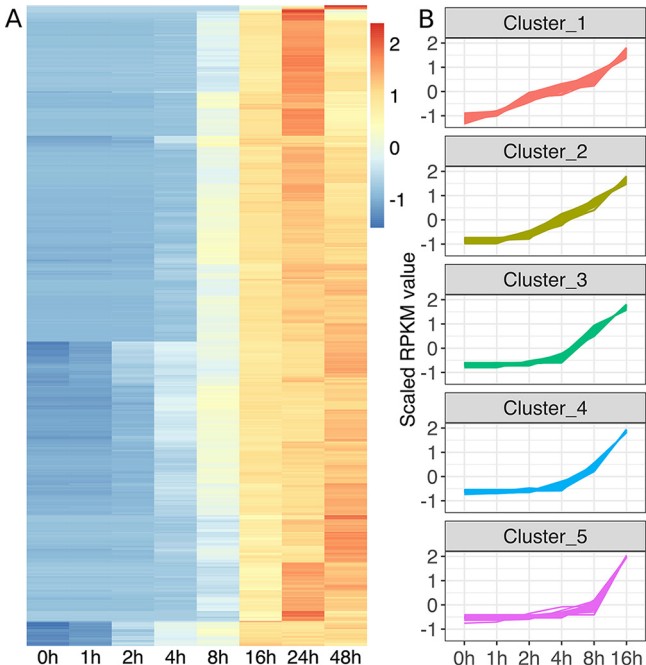

**FIG 2** Expression of medusavirus genes at different time points during the course of medusavirus infection. (A) Heatmap of medusavirus gene expression profiles. Each column represents one time point; each row represents a viral gene; the color scale indicates Z-score scaled RPKM values. (B) Medusavirus temporal gene expression profiles in the five clusters. *x* axis, time points postinfection; *y* axis, Z-score scaled RPKM value for each gene. Each line represents a viral gene.

processing" category (e.g., putative VLTF-2 transcription factor, putative late transcription factor 3, and transcription elongation factor S-II), "virion structure" (e.g., major capsid protein and putative membrane protein), and "translation" (e.g., translation initiation factor eIF1 and a tRNA$^{His}$ guanylyltransferase). Clusters 4 and 5 also contained genes under various functional categories, including those related to transcription, translation, and virion structure, but many genes in these clusters were functionally unannotated (Fig. 3A). Our data indicate relatively late transcription of the 80 genes that encode proteins that are known to be packaged in viral particles (Fig. 3B), and most of them (73 genes, 92%) were in the intermediate or late expression clusters (i.e., clusters 3 to 5) (2).

**Subcellular location of viral gene products.** A large majority of viral gene products were predicted to be transported to the nucleus (131 genes, 28.4%), cytoplasm (170 genes, 36.9%), mitochondrion (51 genes, 11.1%), or extracellular components (37 genes, 8.0%) (Fig. 4). We combined this subcellular localization information with previously identified clusters. The proportion of nucleus-localized proteins showed a clear descending trend in the order of expression clusters, with the highest proportion in cluster 1 (45.2%) and lower proportions (19.7% to 32.6%) in other clusters. The proportion of nucleus-localized proteins in the virion-packaged group (i.e., proteins that are known to be packaged inside the virion) was 21.3%. The proportion of cytoplasm-localized proteins increased from cluster 1 (28.6%) to cluster 3 (43.7%) and then decreased to cluster 5 (20.7%). The proportion of cytoplasm-localized proteins in the virion-packaged group was high (38.8%) and was the biggest category among the virion-packaged proteins. The proportions of mitochondrion- and extracellular-localized proteins increased in cluster 4 (extracellular, 9.5%; mitochondrion, 16.6%) and cluster 5 (extracellular, 13.8%; mitochondrion, 17.2%); however, their proportions in the virion-packaged group was low (5 genes, 6.3% of the total virion proteins). Few proteins were predicted to localize to the cell membrane (22 genes), endoplasmic reticulum (16 genes), peroxisome (12 genes), Golgi apparatus (6 genes), and lysosome/vacuole (4 genes).

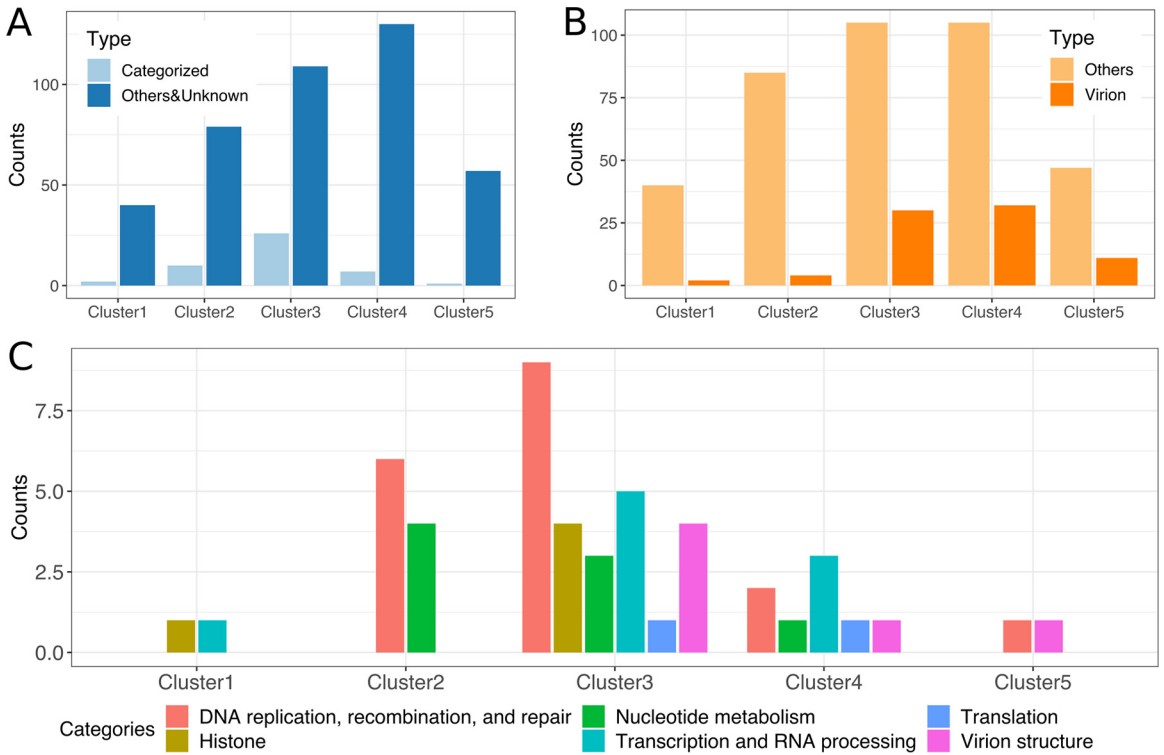

**FIG 3** Distribution of genes with annotated functions among the five expression clusters. (A) Numbers of unknown and annotated genes in the expression clusters. Light blue indicates genes with categorized functions; dark blue indicates genes with uncategorized or unknown function. (B) Numbers of genes in the expression clusters. Light orange indicates genes that encode proteins not packaged in virions; dark orange indicates genes that encode proteins packaged in virions (2). (C) Numbers of functionally annotated genes in each of the expression clusters.

Their proportions and absolute numbers increased in the intermediate or late expression clusters (i.e., clusters 3 to 5) (Data set S6). In addition, half of the peroxisome-localized (6 out of 12 genes) and cell membrane-localized (11 out of 22 genes) proteins were in the virion-packaged groups.

**Putative regulatory elements.** To investigate the regulatory mechanisms of medusavirus gene expression, we analyzed the genomic localizations of the temporal gene expression clusters and associated gene functions. However, this analysis did not detect any definitive features related to the organization of genes in the genome and their temporal or functional groups (Fig. 5). *De novo* motif searches in the 5′ region upstream of the viral genes previously identified two motifs, a palindromic motif (GCCATRTGAVKTCATRTGGYSRSG, 53 occurrences) and a poly-A motif (VMAAMAAMARMAAMA, 251 occurrences) (19). We used the same method and found 3 additional putative promoter motifs, which were statistically significantly overrepresented in the analyzed sequences (E value, $<1 \times 10^{-5}$)—GCCRYCGYCGH (GC-rich motif, 134 occurrences), NRAAWAAA (AATAAA-like motif, 123 occurrences), and GTGTKKGTGGTGGTG (GT-rich motif, 37 occurrences) (Fig. 6; Fig. S2; Tables S1 to S3). In the following paragraphs, we investigate these five motifs with respect to their genomic locations and associations with the expression clusters.

The palindromic motif was preferentially found in the region approximately 40 to 70 bp upstream of the start codon. The poly-A motif was preferentially found in the region approximately 0 to 40 bp upstream of the start codon. The GC-rich motif had no obvious preferred position upstream of the start codon, but it often overlapped upstream genes. The AATAAA motif was preferentially found in the region approximately 0 to 60 bp upstream of the start codon, which is similar to the preferred positions of the poly-A motif. The GT-rich motif was preferentially found close to the start codon (Fig. 6).

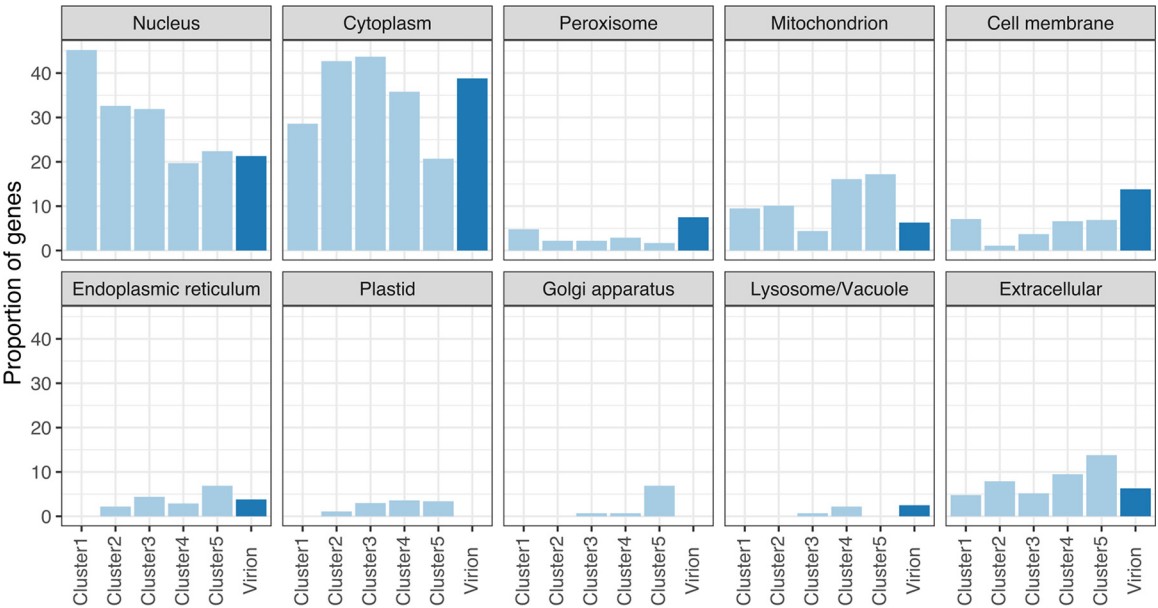

**FIG 4** Predicted subcellular locations of the products of viral genes. The height of each bar indicates the proportion of genes in each cluster. Light blue indicates the proportion of genes in the expression clusters; dark blue indicates the proportion of genes among the viral genes whose products are known to be packaged inside the virion. Amoebas do not possess plastids, but the plastid predictions were retained (see Materials and Methods).

The palindromic motif was highly associated with the genes in cluster 1 (immediate early) (Fig. 7). Among the 53 viral genes with the upstream palindromic motif, 40 (75%) were in cluster 1, and they made up 95% of the genes in this cluster. The 13 other genes with the palindromic motif were distributed among other clusters, 8 in cluster 2 and 5 in clusters 3, 4, and 5 (Fig. 7). Furthermore, among the 56 genes detected at 1 hpi, 49 (88%) had the palindromic motif. We also found that 81.0% of the genes in cluster 1 and 69.7% of the genes in cluster 2 had the upstream poly-A motif, whereas they

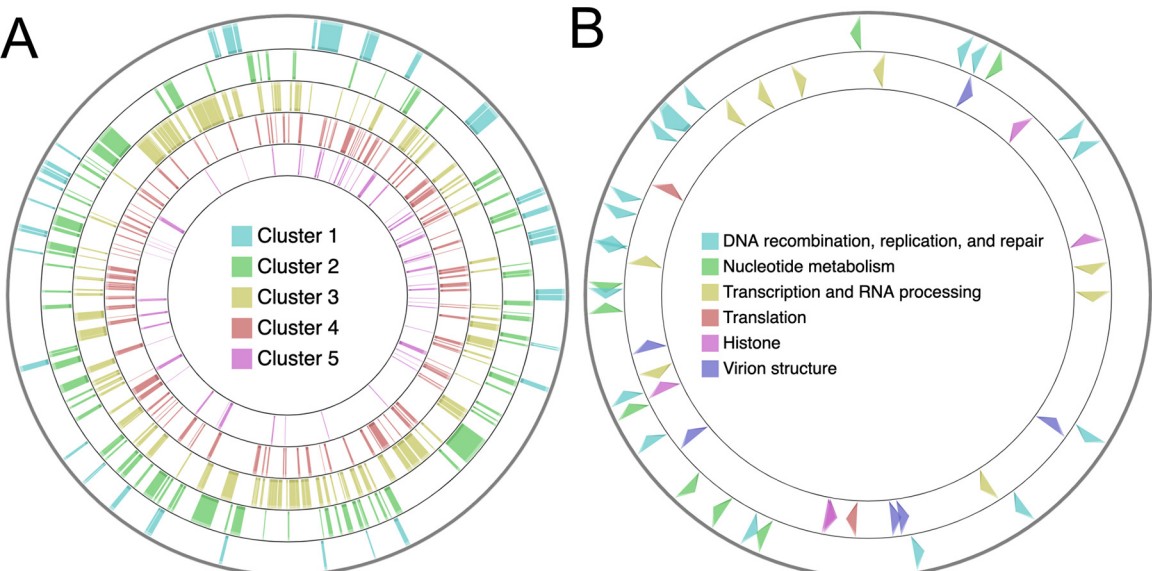

**FIG 5** Organization of genes in the medusavirus genome. (A) Organization of the five expression clusters on the viral genome. (B) Organization of functional groups of genes on the viral genome. (Outside layer) Genes classified in the "DNA replication, recombination, and repair" and "nucleotide metabolism" categories. (Inside layer) Genes classified in the "transcription and RNA processing," "translation," "histones," and "virion structure" categories.

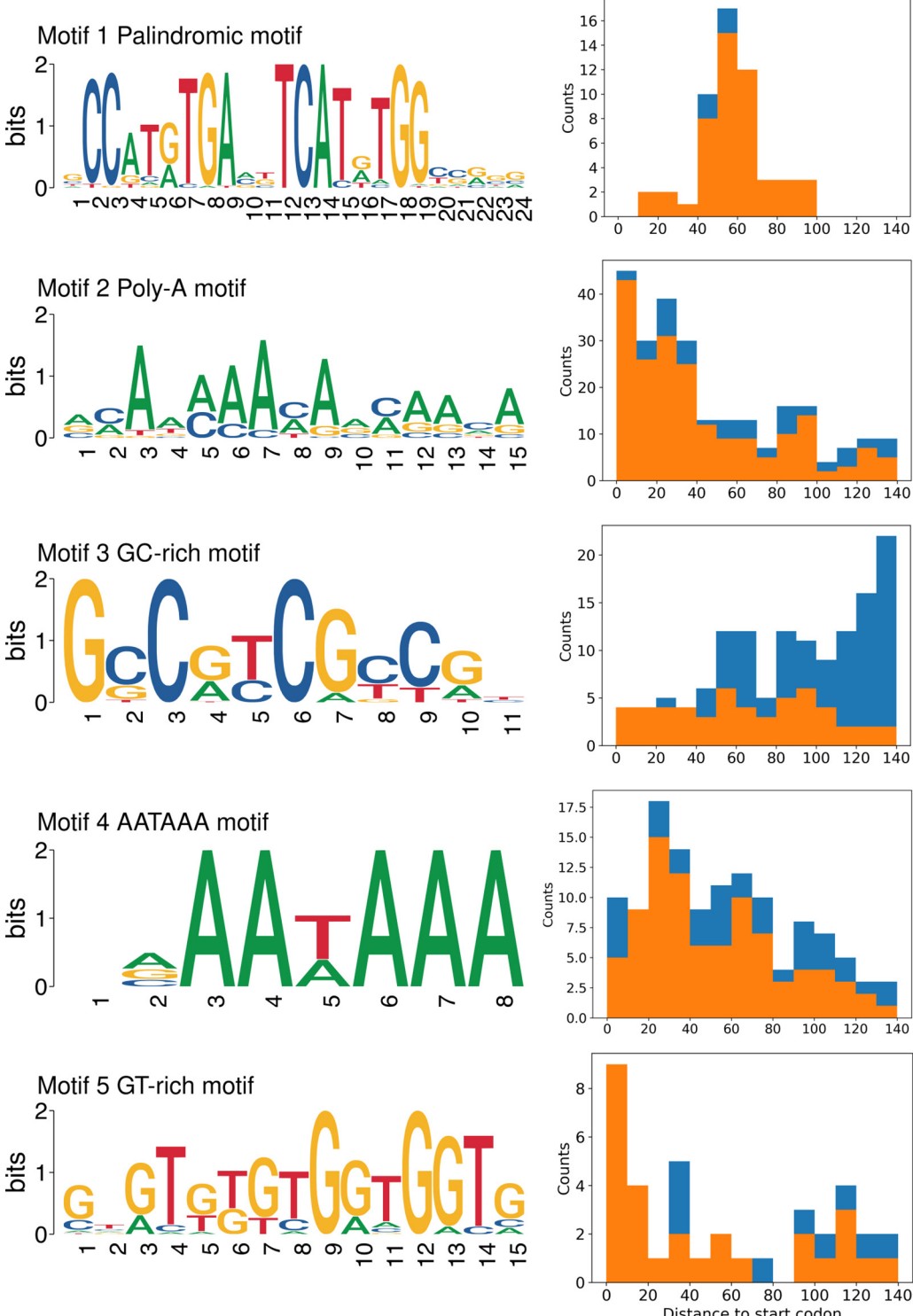

**FIG 6** Sequence motifs enriched in the 5′ region upstream of the genes in the medusavirus genome and their distribution relative to the corresponding start codon. (Left panel) Motif name and its logo; (right panel) distance to the corresponding start codon; orange indicates motifs that did not overlap neighboring genes; blue indicates motifs that overlapped with neighboring genes.

made up only 40.2% to 55.2% of the genes in clusters 3 to 5 (Fig. 7). For the other three motifs, we found no specific association with gene clusters.

To investigate if these upstream motifs were promoter motifs, we scanned the medusavirus genome with these motifs. The palindromic and poly-A motifs were statistically

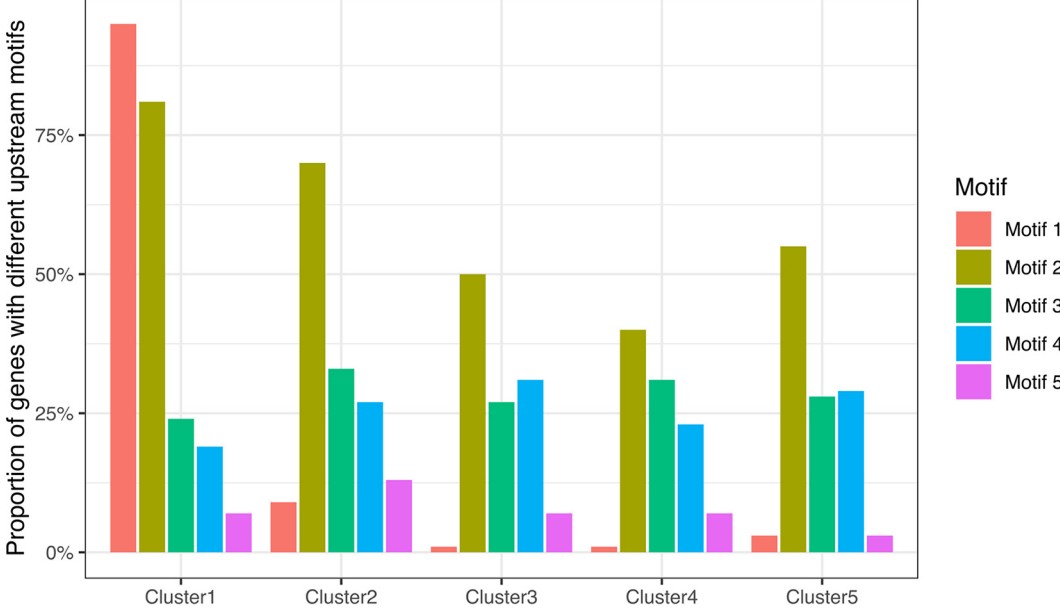

**FIG 7** Proportion of genes with the different upstream motifs in each expression cluster.

significantly more abundant in intergenic regions than in coding sequences ($P <$ $2.2 \times 10^{-16}$; Table 1). Furthermore, these two motifs were more frequent in the upstream intergenic regions of genes than in the downstream intergenic regions ($P <$ $10^{-5}$; Table 2). The other three motifs showed no preference for either intergenic regions or coding sequences, leaving the putative promoter status of these motifs unclear; they may have other functional or structural roles. We also searched the 3′ downstream regions of the medusavirus genes for hairpin structures but failed to identify any. This is different from the presence of hairpin structures in *Acanthamoeba polyphaga mimivirus* (stem length, ≥13 bp; loop, ≤5 bp), *Megavirus chilensis* (stem length, ≥15 bp), *Pithovirus sibericum* (stem length, ≥10 bp; loop, ≤10 bp), and virophage sputnik, noumeavirus, and melbournevirus in the family *Marseilleviridae* (3, 5, 22, 29, 30).

**The host nuclear transcriptional profile was greatly altered.** The proportion of host mRNA reads and their expression levels assessed by RPKM did not show large changes until 8 hpi (Fig. 1 and 8A). After 8 hpi, the proportion of host reads decreased rapidly, and the proportion of viral reads increased. Our cluster analysis of the data set of 0 to 16 hpi showed that the transcription profile of the host *A. castellanii* genes changed greatly between 8 hpi and 16 hpi, with two expression clusters for the host genes (Fig. 9A). Of the 10,627 *A. castellanii* genes examined, 7,970 (75%) were in cluster 1. Their relative expression levels decreased across time, especially between 8 hpi and 16 hpi. The remaining 2,657 (25%) genes were in cluster 2, and their relative expression levels increased (at 16 hpi, mean $\log_2$ fold changes were −0.445 and 0.364 for cluster 1 and cluster 2, respectively).

**TABLE 1** Distribution of upstream motifs in intergenic regions[a] of the medusavirus genome

| | Data for: | | | | |
|---|---|---|---|---|---|
| | Motif 1 | Motif 2 | Motif 3 | Motif 4 | Motif 5 |
| Total count | 48 | 152 | 807 | 19 | 253 |
| Count in IR[b] | 30 | 84 | 43 | 19 | 27 |
| Background frequency[c] | 0.105 | | | | |
| *P* value | 4.60e-18 | 5.62e-42 | 1.00 | 0.132 | 0.495 |

[a]A binomial test was used to assess whether each motif was preferentially located in intergenic regions (IRs).
[b]Only motifs that did not overlap predicted genes were considered to be located in IRs.
[c]Background frequency was calculated by dividing the sum of the length of all IRs by the length of the whole genome.

**TABLE 2** Preference of the palindromic and poly-A motifs for up- or downstream regions of medusavirus genes[a]

| | Motif 1 palindromic | | Motif 2 Poly-A | |
|---|---|---|---|---|
| | **With motif** | **Without motif** | **With motif** | **Without motif** |
| Divergent[b] | 22 | 93 | 40 | 75 |
| Convergent[c] | 0 | 88 | 4 | 84 |
| P value | 2.76e-06 | | 8.48e-08 | |

[a]Only motifs predicted to be located in intergenic regions were used to determine their preferred location. The P values were calculated by the Fisher exact test.
[b]Divergent cases were defined as motifs being located in the upstream regions of both neighbor genes.
[c]The convergent cases were defined as the motifs being located in the downstream region of both neighbor genes.

Based on the host gene clusters, we performed Gene Ontology GO and KEGG pathway functional enrichment analyses (Fig. 9B to E). Cluster 1 genes were enriched in cellular transportation-related GO terms, such as "localization," "establishment of localization," and "transport" (Fig. 9B). Cluster 2 genes were enriched in 60 GO terms that fell into two main categories (Fig. 9D). One category comprised terms related to "cellular protein metabolic process" and "proteolysis involved in cellular protein catabolic process," and the other category comprised stress-related terms such as "DNA repair" (Fig. S3).

Cluster 1 host genes were also enriched in KEGG pathways, including "biosynthesis of secondary metabolites," "glycerophospholipid metabolism," "aminoacyl-tRNA biosynthesis," "glycerolipid metabolism" and "ABC transporters" (Fig. 9C). Cluster 2 host genes were enriched with "Ribosome," "Ubiquitin mediated proteolysis," "Proteasome," "Base excision repair," "Homologous recombination," and "Nucleotide excision repair" (Fig. 9D; Fig. S4). Clearly, the first three of these pathways correspond to the enriched GO terms in cluster 2 related to protein catabolic and metabolic processes, and the latter three pathways correspond to the enriched GO terms related to DNA repair and stress response.

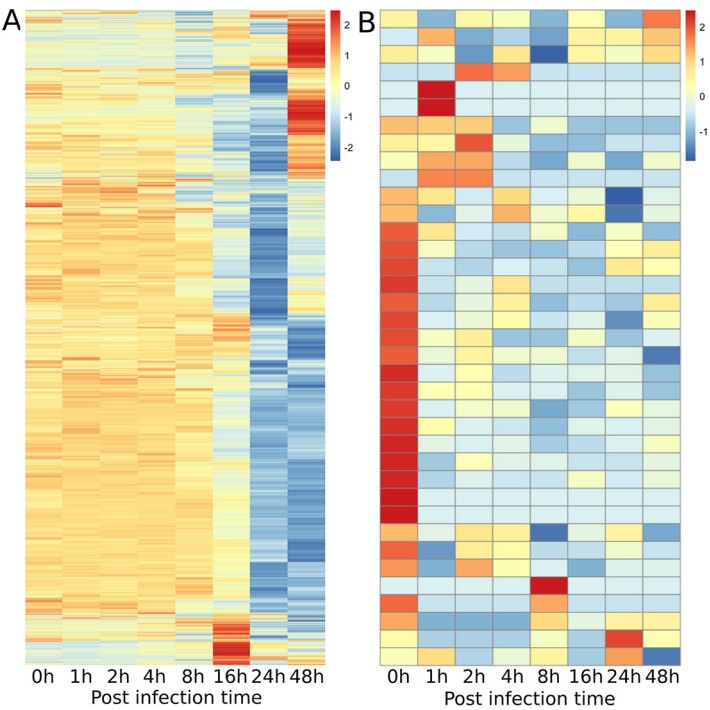

**FIG 8** Transcription profiles of the *Acanthamoeba castellanii* (host) genes. (A) Host nuclear genes. (B) Host mitochondrial genes. x axis, time points of the infection cycle; y axis, different genes in the host genome and its mitochondrial genome. The color scale indicates Z-score scaled RPKM values.

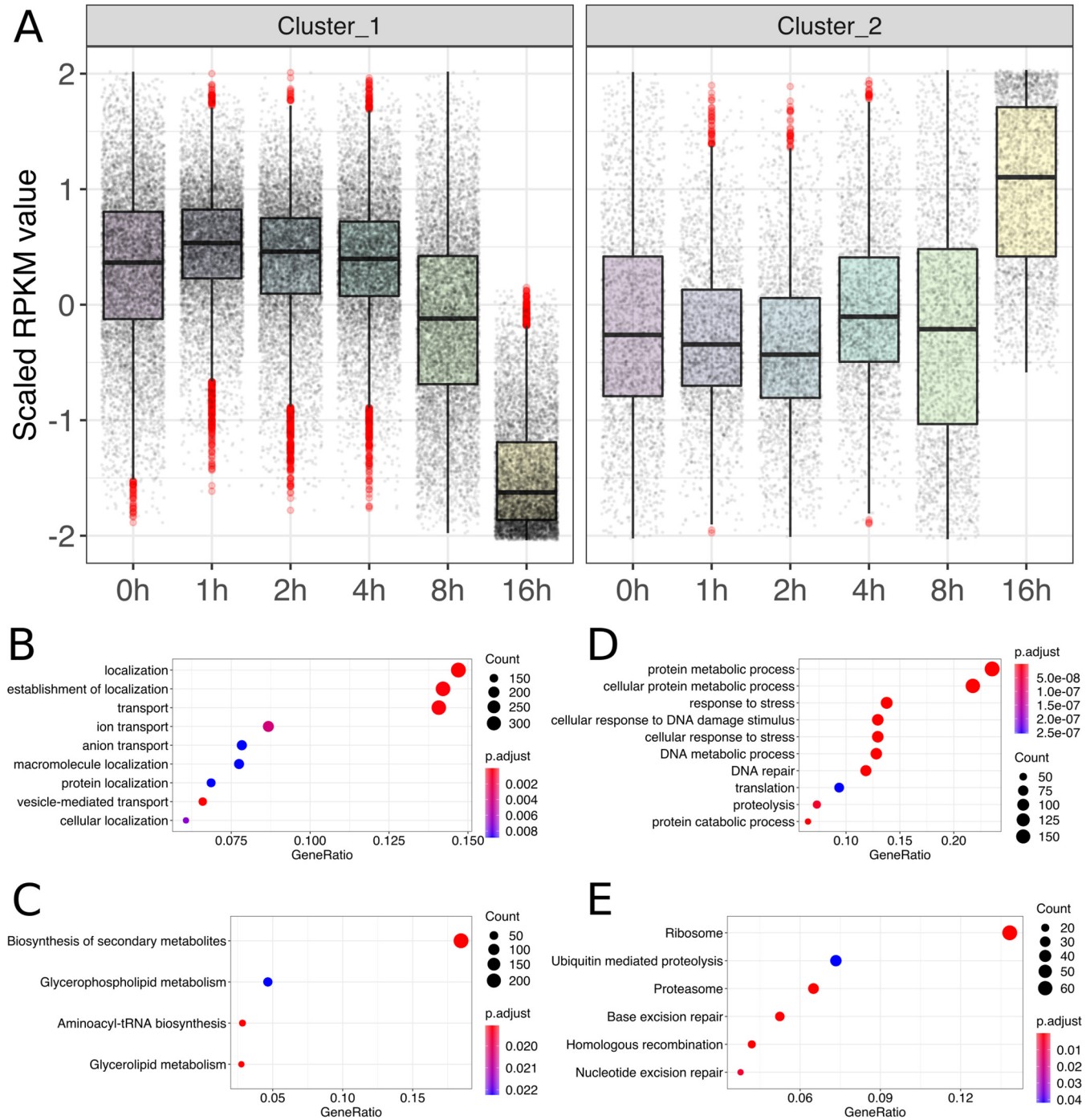

FIG 9 *Acanthamoeba castellanii* (host) nuclear gene expression clusters and their predicted functions. (A) Two expression clusters were identified for the host nuclear genes. (B) Enriched GO terms for the genes in cluster 1. (C) Enriched KEGG pathways for the genes in cluster 1. (D) Enriched GO terms for the genes in cluster 2. (E) Enriched KEGG pathways for the genes in cluster 2.

**The host nuclear gene expression pattern changed after 16 hpi.** Clusters identified in the first 16 hpi did not maintain their expression patterns after 16 hpi (Fig. S5). The expression levels of some genes annotated with the GO term "transport" were increased greatly at 48 hpi. In contrast, cluster 2 genes, which were activated at 16 hpi, were suppressed at 24 hpi and then recovered to some extent at 48 hpi. We found that some of the genes that were activated at 48 hpi were encystment-mediating genes, which included an encystation-mediating serine proteinase (EMSP), eight

cysteine protease proteins, cyst-specific protein 21 (CSP21), and two cellulose syn-thases (31–35) (Fig. S6).

**Mitochondrial expression was maintained during medusavirus infection.** In our RNA-seq data, 37 of the 53 genes encoded in the *A. castellanii* mitochondrial genome had at least one read count during the course of the infection (Fig. 8B). The numbers of mitochondrial mRNA reads were low (4,365 reads at 0 hpi and maintained at <2,500 at the later time points). However, the numbers of mitochondrial reads were more sta-ble than the numbers of reads that were mapped onto the nuclear genome, the latter of which decreased by about 30% during the course of the infection.

Mitochondrial genes with the highest read counts were related to rRNA genes (AccaoMp41, 23S-like rRNA and AccaoMp42, 16S-like rRNA), followed by energy metab-olism [AccaoMp13, H (+)-transporting ATPase subunit 9]. These genes showed a sud-den decrease at 1 hpi (~2-fold decrease in RPKM) but then maintained at later time points. The proportions of the other mapped mitochondrial mRNA reads were small. Among the 16 mitochondrial genes that were not detected in the RNA-seq data, 11 were tRNA genes, 3 were ribosomal protein genes, and 2 genes were of unknown function.

## DISCUSSION

We performed RNA-seq to dissect the transcriptional program of medusavirus. Medusavirus has been reported to initiate its genome replication in the host nucleus and maintain the nuclear membrane intact during the infection cycle, with occasional induction of the encystment of the host amoeba *A. castellanii* at approximately 48 hpi (2). We found that transcription began in less than 1 h after the start of infection even though medusavirus has no RNA polymerase genes. Compared with other amoeba-infecting viruses, the speed of the medusavirus infection cycle was slow and appa-rently weak, because it took approximately 24 h for the virus to reach its expression peak, which was still only about 35% of the total mRNAs (at a multiplicity of infection [MOI] of 2.88). In contrast, mimivirus and marseillevirus genes occupy 80% of the total mRNA library at less than 6 hpi (at an MOI of 100 and 1,000 for marseillevirus and mim-ivirus, respectively) (25, 26). The slow and mild medusavirus infection may be explained by the different MOI used in the infection experiments, as a higher MOI has been reported to accelerate the infection course (36, 37). An additional explanation may be a slow start of medusavirus replication. Mimivirus carries its RNA polymerase in the vi-ral particles to initiate its transcriptional process as soon as the viral particles open up in the cytoplasm (37). In contrast, medusavirus encodes no RNA polymerase and thus depends largely on host transcriptional machinery, which may account for its slow infection.

Clustering of medusavirus gene expression profiles showed clear temporal expres-sion patterns akin to those observed for other giant viruses (25, 26, 38, 39). Of isolated viruses, medusaviruses are the only viruses that encode the linker histone in addition to the core histone domains (2, 19). Therefore, the functional relationship between the linker histone H1 and the core histones was a focus of this study. We found that linker histone H1, which is not packaged in viral particles (2), was transcribed immediately af-ter the beginning of transcription. In contrast, the four core histones, which are carried in virions (2), started to be transcribed later. The different transcriptional profiles between the linker histone H1 and core histones suggest different functional roles between them. Histone H1 may cooperate with high-mobility group proteins in viral particles to regulate the accessibility of the viral genome for the subsequent transcrip-tion process (40, 41), or it may function to regulate the host chromatin. Regarding viral core histone proteins, the core histone proteins of marseilleviruses have been shown to bind DNA and form a structure resembling eukaryotic nucleosomes (42, 43). Marseillevirus histones have been also shown to localize the cytoplasmic viral factories and mature virions in the end of infection (43). Medusavirus core histones may func-tion in a similar way for viral genome packaging as in marseilleviruses.

The predicted subcellular localization of viral gene products showed that cluster 1 had a higher proportion of nucleus-localized proteins than the other clusters. The predicted proportions of nucleus-localized proteins in the medusavirus and medusavirus stheno genomes were ranked 7th and 4th, respectively, among all known amoeba-infecting NCLDVs, indicating the importance of remodeling the nuclear environment immediately after medusavirus infection (Fig. S1). The remodeling probably contributes to subsequent viral gene transcription and DNA replication within the host nucleus. Putative cytoplasm-localized proteins were enriched in the virion-packaged group (31 genes, 38.8% of genes in virions), and almost half of the cell membrane and peroxisome-localized proteins were also packaged inside virions, suggesting that there may be interactions between virion-packaged proteins and the host cytoplasm and other subcellular membrane-bound compartments at an early phase of infection. The increasing expression of genes targeting the mitochondrion, endoplasmic reticulum, and Golgi apparatus suggests that medusavirus synthesizes these genes, probably to maintain or reprogram the functions of these organelles, after starting infection, rather than bringing them within the virion.

The enrichment of the palindromic motif in the upstream region of genes that were transcribed immediately after infection suggests that this motif may be the immediate early promoter of medusavirus genes. The poly-A motif that we detected in the upstream region of early expressed genes is reminiscent of the A/T-rich early promoter motifs found in other giant viruses in the phylum *Nucleocytoviricota* that have been proposed to have a common ancestral promoter motif, TATATAAAATTGA (44–47). The poly-A motif in the upstream regions of the medusavirus genes may have evolved from this common ancestral motif. Although the AATAAA motif was not preferentially located in the intergenic regions, it is similar to the 3′-end motif in the polyadenylation signal sequence in eukaryotes (48). The AATAAA motif also was detected in mimivirus, but it did not function as a polyadenylation signal (29). Regarding the 3′-end processing mechanism of giant viruses, A/T-rich hairpin structures have been identified after stop codons (3, 5, 22, 29, 30), and proteins that can recognize these structure have been studied (49). However, we did not find any A/T-rich hairpin structures in the 3′ downstream regions of medusavirus genes.

We identified two temporal clusters for host genes during viral replication. The fact that a majority (75%) of host genes showed decreases in their relative expression level at 16 hpi suggests that the host genes experienced global suppression. GO terms related to localization and transport were enriched in host cluster 1, suggesting that decreased transport activity occurred within the host cell during the course of the virus infection. In addition, the increased representation of the KEGG pathways "ribosome" and "proteosome" and the GO terms "cellular protein metabolic process" and "proteolysis involved in cellular protein catabolic process" suggests an increased activity of viral protein synthesis and degradation of host proteins, which needs experimental validation. We found enriched homologous recombination and DNA repair related GO and KEGG terms at 16 hpi (Fig. 9C and D; Fig. S4). Their increased representation, which has been reported to aid polyomavirus reproduction (simian virus 40 and JC polyomavirus) (50–52), may actively help medusavirus reproduction, although it may be due to a host response against virus infection. We also found an overrepresentation of encystment-related genes at 48 hpi (Fig. S6). As the culture may be a mixture of infected and uninfected amoeba cells at this time point with the initial MOI of 2.88, determining the cause of this overrepresentation (i.e., due to either healthy or infected cells) requires further investigation. Of note, encystment of both infected and healthy *Veramoeba vermiformis* cells has been observed upon infection by Faustovirus meriensis and has been suggested as an antiviral mechanism of the host trapping the viruses inside the cyst walls (53). A similar host strategy may be working for the *A. castellanii*-medusavirus infection system.

The expression pattern of the *A. castellanii* mitochondrial genes during the course of medusavirus infection was similar to the pattern found in marseillevirus (26). All tRNA-encoding genes had low expression levels, possibly because transcripts with a poly-A tail were used to build the RNA library and tRNA genes do not have poly-A tails. The genes with the highest expression levels included genes involved in energy metabolism and rRNA genes. Unlike the host nuclear genes, the transcriptional activity of these mitochondrial genes was maintained after 1 hpi, suggesting that host mitochondria may stably supply energy for viral replication.

In summary, our transcriptome data clearly delineated five temporal expression clusters for viral genes. Most of the immediate early genes (cluster 1) were of unknown function and had a palindromic promoter-like motif upstream of their start codons. Many of the immediate early gene products were predicted to localize in the host nucleus, suggesting that medusavirus modifies the host nuclear environment right after the start of infection by involving the action of dozens of viral genes. The genes that were expressed later (clusters 2 to 5) have various functions. The viral histone H1 gene is in the cluster 1, whereas the four core histone genes are in cluster 3, suggesting that they have distinct roles in viral replication. The transcriptional landscape of host nuclear genes was altered during infection, especially after 8 hpi. At 16 hpi, the host nuclear transcription showed a great alteration. Our transcriptome data will serve as a fundamental resource for further investigation of the infection strategies of medusaviruses, which are a group of amoeba-infecting giant viruses that have no close relatives among the diverse NCLDVs.

## MATERIALS AND METHODS

**Amoeba culture, virus infection, and sequencing.** *Acanthamoeba castellanii* strain Neff (ATCC 30010) cells were purchased from the American Type Culture Collection (ATCC; Manassas, VA, USA). The *A. castellanii* cells were cultured in eight 75-cm$^2$ flasks with 25 ml of peptone-yeast-glucose (PYG) medium at 26°C for 1 h and then infected with purified medusavirus as previously described (2), at a multiplicity of infection (MOI) of 2.88. The titer of medusavirus was measured by 50% tissue culture infective dose (TCID$_{50}$) by inoculating fresh amoeba solution on a 96-well plate with a serially diluted virus solution (54). In a previous study, infection of medusavirus was associated with the appearance of the host amoebas forming cysts at an MOI of about 1 to 2 (2). With the aim of investigating this phenomenon, we performed our infection experiment with a similar MOI level. After addition of medusavirus to 7 of the 8 flasks (1 was the negative control), cells were harvested from each flask at 1, 2, 4, 8, 16, 24, and 48 hpi. Each cell pellet was washed with 1 ml of phosphate-buffered saline (PBS) by centrifugation (500 × *g*, 5 min at room temperature). Total RNA extraction was performed with an RNeasy minikit (Qiagen, Inc., Japan) and quality checked by agarose gel electrophoresis. The extracted RNA was sent to Macrogen Corp., Japan, for cDNA synthesis and library construction.

The cDNA synthesis and library construction were done using a TruSeq stranded mRNA low-throughput (LT) sample prep kit (Illumina, Inc., San Diego, CA, USA) following the manufacturer's protocol. Briefly, the poly-A-containing mRNAs were purified using poly-T oligonucleotide attached magnetic beads. Then, the mRNA was fragmented using divalent cations under elevated temperature. First-strand cDNA was obtained using reverse transcriptase and random primers. After second-strand synthesis, the cDNAs were adenylated at their 3′ ends, and adaptors were added. The DNA fragments were amplified by PCR and purified to create the final cDNA library. The RNA-seq was performed on a NovaSeq 6000 platform (Illumina, Inc.).

**Read mapping and count normalization.** The quality of the obtained reads was checked using the FastQC tool (http://www.bioinformatics.babraham.ac.uk/projects/fastqc/), which showed that the overall quality was above the threshold (quality threshold, ≥20; no known adapters). Thus, we did no further trimming of the reads. The mRNA reads were mapped to a merged data set composed of the nuclear genome of *A. castellanii* (GCF_000313135.1_Acastellanii.strNEFF_v1), the medusavirus genome (GenBank accession number AP018495.1), and the mitochondria genome of *A. castellanii* (GenBank accession number NC_001637.1) using HISAT 2 (55) with a maximum intron size of 1,000 bp. The number of reads mapped on each gene was calculated using HTSeq in union mode (56). The transcriptional activity of genes was estimated by reads per kilobases of transcript per million mapped reads (RPKM) (Data sets S1 to 4).

**Clustering.** To discover the transcriptional patterns during medusavirus infection, we clustered the transcription profiles of viral and amoeba nuclear genes using the *k*-means method. We chose the library from 0 to 16 hpi to cluster viral genes, because a previous study indicated that replicated viral DNA was first observed in the cytoplasm at approximately 14 hpi and new virions were also observed to be released at the same time point (2), which indicated the termination of a cycle of infection at this time point. Genes with at least one mapped read across the 0 to 16 hpi libraries were included in the downstream analysis. To define the optimal number of clusters without prior biological information, we used the R packages NbClust and clusterCrit, which use different clustering indices to estimate the quality of clusters (57, 58). For virus genes, most indices gave 5 as the optimal number of clusters, and for amoeba

nuclear genes, most indices gave 2 as the optimal number of clusters. Therefore, we performed the *k*-means clustering with $k = 5$ and 2 for viral and amoeba nuclear genes, respectively (Data set S5). We did not perform clustering of the expression of mitochondrial genes but analyzed expression of individual genes based on RPKM values.

**Subcellular localization prediction of viral genes.** Subcellular localization prediction of medusavirus genes was performed using DeepLoc 1.0 (59). We also predicted the subcellular localization tendency of other amoeba-infecting NCLDVs using the same method (Data sets S6 and S7; Fig. S1). A minor proportion of genes (0.0 to 5.0% for each virus) were predicted to target the plastid. Although amoebas do not possess plastids, we kept these predictions as they are, because even though these viruses were isolated using amoeba coculture, there remains a possibility that their natural hosts possess plastids.

**Sequence motif analysis.** MEME 5.1.1 was used for *de novo* motif prediction in the 5′ upstream sequence of medusavirus (60). We extracted 150-bp sequences upstream of the open reading frames. MEME was used in classic mode with motif width ranges of 8 to 10 bp, 6 to 15 bp, and 8 to 25 bp and "zero or more motifs in each intergenic region." We adopted the results with the motif width range of 8 to 25 bp because this was the only range in which the palindromic motif was detected (Fig. S2; Tables S1 to S3). We used the FIMO software tool (61) to scan the medusavirus genome for motifs that were predicted by MEME. The RNAMotif 3.1.1 algorithm (62) was used to find A/T-rich hairpin structures in the region downstream of each stop codon in medusavirus genes.

**Functional enrichment analysis.** Gene ontology (GO) and KEGG pathway enrichment analysis of the identified clusters of host genes were performed using the ClusterProfiler package in R (63).

**Data availability.** The sequencing data used in this study have been submitted to the DDBJ under the accession number DRA011802.

## SUPPLEMENTAL MATERIAL

Supplemental material is available online only.
**SUPPLEMENTAL FILE 1**, XLSX file, 1.8 MB.
**SUPPLEMENTAL FILE 2**, PDF file, 0.04 MB.
**SUPPLEMENTAL FILE 3**, PDF file, 1 MB.

## ACKNOWLEDGMENTS

We thank Hiroyuki Hikida for helpful discussion in revising our manuscript. Computational time was provided by the SuperComputer System, Institute for Chemical Research, Kyoto University. This work was supported by JSPS/KAKENHI (no. 18H02279 and 20H03078), The Kyoto University Foundation, and the International Collaborative Research Program of the Institute for Chemical Research, Kyoto University (no. 2018-32, 2019-34, and 2020-31).

We thank Margaret Biswas, Ph.D., from Edanz Group for editing a draft of the manuscript.

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
