## [Reviewer comments · Microbiology Spectrum]

Microbiology Spectrum

RNA-seq of medusavirus suggests remodeling of the host nuclear environment at an early infection stage

Ruixuan Zhang, Hisashi Endo, Masaharu Takemura, and Hiroyuki Ogata

Corresponding Author(s): Hiroyuki Ogata, Kyoto University

Review Timeline:

Submission Date:	April 23, 2021
Editorial Decision:	June 14, 2021
Revision Received:	August 12, 2021
Accepted:	August 16, 2021

Editor: Samuel Campos

Reviewer(s): The reviewers have opted to remain anonymous.

Transaction Report:

DOI: <https://doi.org/10.1128/Spectrum.00064-21>

June 14, 2021

Prof. Hiroyuki Ogata
Kyoto University
Institute for Chemical Research
Uji, Kyoto 611-0011
Japan

Re: Spectrum00064-21 (RNA-seq of medusavirus suggests remodeling of the host nuclear environment at an early infection stage)

Dear Prof. Hiroyuki Ogata:

Thank you for submitting your manuscript to Microbiology Spectrum. Your manuscript has been evaluated by three reviewers and the decision is for acceptance, contingent on minor modifications. Please review the reviewers' critiques and respond to their concerns, as outlined below.

When submitting the revised version of your paper, please provide (1) point-by-point responses to the issues raised by the reviewers as file type "Response to Reviewers," not in your cover letter, and (2) a PDF file that indicates the changes from the original submission (by highlighting or underlining the changes) as file type "Marked Up Manuscript - For Review Only". Please use this link to submit your revised manuscript - we strongly recommend that you submit your paper within the next 60 days or reach out to me. Detailed information on submitting your revised paper are below.

Link Not Available

Sincerely,

Samuel Campos

Journals Department
Reviewer comments:

Reviewer #1 (Comments for the Author):

In this study, the authors have investigated the transcriptional profile of medusavirus - Acanthamoeba infection cycle and provided data on the potential regulatory and functional landscapes underpinning the observed transcription patterns. I have provided my comments below:

Regarding data on host-virus infection progression: One of the key data product is not provided. Specifically, there is no data in this paper regarding the proportion and cell counts of healthy and infected cells over the course of the experiment. Similarly, no data on the number of virus particle produced is provided. Given this is a host-pathogen infection experiment, without these data, contextualizing the gene expression result is difficult. For example, the authors indicate that expression of encystment related genes had higher expression around 48 hours, however, which control this expression level was compared to is not provided. If the expression of encystment related genes is as high as what they found at the beginning of the experiment, then it might mean that these expressions are coming from healthy cells - because we don't have any data regarding new virus infections around 24 or 48 hours. The authors should therefor provide the cell and virus count data and if possible, data on the proportion of infected and healthy amoeba cells over the course of the experiment.

Line 67: medusavirus stheno - 'medusavirus' should be capitalized.

Line 117: What was the reason to use the MOI 2.88 and not an MOI of 1 (or other values)? Please specify the rationale. Thanks!

Line 157: Why did the authors choose the 'second-best' prediction removing the plastid prediction? The way I understand, the prediction with highest probability should be chosen. I understand that for amoeba infecting viruses, 'plastid' prediction likely is a false prediction. In that case the best approach should be to avoid/remove these genes from further analysis where plastid signal was predicted -rather than choosing the second best hit. At the very least, the authors should explain this caveat in their methods - and also provide the number/percent of such cases for each genome where they found plastid to be the best match and chose the second best hits instead.

Fig S1: the authors included viruses that not only infect amoeba but other hosts in this analysis (for example tetraselmis virus-1). They should correct this in the manuscript, tables and figures - wherever it is necessary.

Line 190-194: The classification of genes in early, late or intermediate needs to be more specific. For example, the authors say that "cluster 2 ("early"), genes that start to be expressed at 1-2 hpi" This will mean that these genes showed no expression before that time point, and there was no RNA-seq reads mapped on these genes before the specified time point. Is that true? If these genes showed some expression before 1-2 hr, then this statement will be incorrect. Based on the data, the authors might want to use a more general language - for example, early genes showed gradual increase in expression starting at 1 hr.

The statements in these paragraphs should be modified as such - or if the authors have alternative explanations, that should be mentioned.

Reviewer #2 (Comments for the Author):

In their manuscript entitled „RNA-seq of medusavirus suggests remodeling of the host nuclear environment at an early infection stage“, Zhang et al. performed a time course RNA-seq analysis of *Acanthamoeba castellanii* cells infected by medusavirus. The manuscript describes the results in a clear and concise manner and is well-organized with figures highlighting their main findings. In my opinion, the manuscript would benefit in general having some conclusions for each result paragraph and maybe, statements from the authors or hypothesis to be tested in the future. I must state that I am not qualified to assess the technical set-up of their RNA-seq analysis and have some minor comments regarding the manuscript:

1. In the first paragraph of the introduction, there is a confusion between giant viruses and NCLDV's that also include large DNA viruses like poxviruses, asfarviruses, etc. It would help if the authors would refer either to NCLDV's and then giant viruses in particular.
2. Line 63: "cosmopolitan" should be "ubiquitous"
3. In the second paragraph of the introduction, it is unclear to me if the authors refer to replication as the viral replication cycle including genome replication and particle assembly or only to replication of the viral genome. Please, specify.
4. In the materials and methods, the methods for production, purification and titration of the virus should be explained.
5. Lines 146-148: it is unclear to me if the viral DNA is first observed at 14 hpi or only observed in general. Regarding the new virions, are they also starting to be released at 14 hpi? And how are the viral DNA and new virion detections performed?
6. Lines 153-154: could the author give some information on the mitochondrial genes?
7. Lines 191-194: the authors need to comment a bit more on the difference between clusters 3 and 4, the current sentence is quite vague and the separation between these two clusters, which have the same expression profiles, need to be better justified.
8. In Fig. 3A, the legend mentions genes with "known" and "unknown" functions, while in the figure panel, it is referred to as "categorized" and "others/unknown". What is the difference between these categories? It should be consistent.
9. In Fig. 3B, a reference is missing regarding the analysis of proteins found in virions.
10. Line 207: the authors should be more specific about the bacteriophage (which viral family, etc.).
11. Lines 219-239: the word "target" should be replaced by "localized" or "specific" for each cellular compartment.
12. Line 235: what does "vacuole" refer to?
13. In Fig. 4, label the Y axis more specifically, e.g. "Proportion of genes".
14. In Fig. 7, label the Y axis more specifically, e.g. "Proportion of genes with the different upstream motifs".
15. Line 264: should refer to figure 7 not 6, I believe.
16. Line 365: a reference is missing.
17. Line 394: "recognizing" should be "recognized"
18. In general, the idea that the temporal clusters correspond to a classification based on the start of gene expression which is then maintained and continuously increasing up to 16 hpi should be introduced.
19. Line 452: after "suggesting" a "that" is missing.

Reviewer #3 (Comments for the Author):

The manuscript by Zhang R et al. is a good paper about RNA-seq of the fascinating medusavirus that has, contrary to other giant viruses, the capability to enter host nucleus. I have no specific modifications to suggest but believe that authors could improve slightly the

discussion by presenting Marseilleviruses, the family of viruses harboring histone genes and for which two recent papers have revealed the structure (doi:

<https://doi.org/10.1101/2021.04.29.441998>) (<https://doi.org/10.1038/s41594-021-00585-7>)

Staff Comments:

Preparing Revision Guidelines

For complete guidelines on revision requirements, please see the Instructions to Authors at [link to page]. **Submissions of a paper that does not conform to Microbiology Spectrum guidelines will delay acceptance of your manuscript.**

Please return the manuscript within 60 days; if you cannot complete the modification within this time period, please contact me. If you do not wish to modify the manuscript and prefer to submit it to another journal, please notify me of your decision immediately so that the manuscript may be formally withdrawn from consideration by Microbiology Spectrum.

If you would like to submit an image for consideration as the Featured Image for an issue, please contact Spectrum staff.

In their manuscript entitled „RNA-seq of medusavirus suggests remodeling of the host nuclear environment at an early infection stage”, Zhang et al. performed a time course RNA-seq analysis of *Acanthamoeba castellanii* cells infected by medusavirus. The manuscript describes the results in a clear and concise manner and is well-organized with figures highlighting their main findings. In my opinion, the manuscript would benefit in general having some conclusions for each result paragraph and maybe, statements from the authors or hypothesis to be tested in the future.

I must state that I am not qualified to assess the technical set-up of their RNA-seq analysis and have some minor comments regarding the manuscript:

1. In the first paragraph of the introduction, there is a confusion between giant viruses and NCLDVs that also include large DNA viruses like poxviruses, asfarviruses, etc. It would help if the authors would refer either to NCLDVs and then giant viruses in particular.
2. Line 63: “cosmopolitan” should be “ubiquitous”
3. In the second paragraph of the introduction, it is unclear to me if the authors refer to replication as the viral replication cycle including genome replication and particle assembly or only to replication of the viral genome. Please, specify.
4. In the materials and methods, the methods for production, purification and titration of the virus should be explained.
5. Lines 146-148: it is unclear to me if the viral DNA is first observed at 14 hpi or only observed in general. Regarding the new virions, are they also starting to be released at 14 hpi? And how are the viral DNA and new virion detections performed?
6. Lines 153-154: could the author give some information on the mitochondrial genes?
7. Lines 191-194: the authors need to comment a bit more on the difference between clusters 3 and 4, the current sentence is quite vague and the separation between these two clusters, which have the same expression profiles, need to be better justified.
8. In Fig. 3A, the legend mentions genes with “known” and “unknown” functions, while in the figure panel, it is referred to as “categorized” and “others/unknown”. What is the difference between these categories? It should be consistent.
9. In Fig. 3B, a reference is missing regarding the analysis of proteins found in virions.
10. Line 207: the authors should be more specific about the bacteriophage (which viral family, etc.).
11. Lines 219-239: the word “target” should be replaced by “localized” or “specific” for each cellular compartment.
12. Line 235: what does “vacuole” refer to?
13. In Fig. 4, label the Y axis more specifically, e.g. “Proportion of genes”.
14. In Fig. 7, label the Y axis more specifically, e.g. “Proportion of genes with the different upstream motifs”.
15. Line 264: should refer to figure 7 not 6, I believe.
16. Line 365: a reference is missing.
17. Line 394: “recognizing” should be “recognized”
18. In general, the idea that the temporal clusters correspond to a classification based on the start of gene expression which is then maintained and continuously increasing up to 16 hpi should be introduced.
19. Line 452: after “suggesting” a “that” is missing.

Reviewer comments:

Reviewer #1 (Comments for the Author):

In this study, the authors have investigated the transcriptional profile of medusavirus - Acanthamoeba infection cycle and provided data on the potential regulatory and functional landscapes underpinning the observed transcription patterns. I have provided my comments below:

Regarding data on host-virus infection progression: One of the key data product is not provided. Specifically, there is no data in this paper regarding the proportion and cell counts of healthy and infected cells over the course of the experiment. Similarly, no data on the number of virus particle produced is provided. Given this is a host-pathogen infection experiment, without these data, contextualizing the gene expression result is difficult. For example, the authors indicate that expression of encystment related genes had higher expression around 48 hours, however, which control this expression level was compared to is not provided. If the expression of encystment related genes is as high as what they found at the beginning of the experiment, then it might mean that these expressions are coming from healthy cells - because we don't have any data regarding new virus infections around 24 or 48 hours. The authors should therefor provide the cell and virus count data and if possible, data on the proportion of infected and healthy amoeba cells over the course of the experiment.

Thank you for your comment. We infected the host amoeba at $MOI = 2.88$. MOI alone already indicates the ratio of infection in theory. At $MOI=2.88$, 90% of cells are expected to be infected at the first round of infection. Therefore, as you point out, the total RNA library would be derived from a mixture of healthy and infected amoebas even near the end of our infection experiment. Therefore, we acknowledge that it is impossible from our experimental data to attribute the dynamics of host transcriptome exclusively to either healthy or infected amoebas near the end of experiment (>24hpi).

Regarding the encystment related genes you mention, we agree with you. When we state that these genes are over-represented at 48 hpi, the relative abundances of these genes within the whole host transcripts were compared with their relative abundance at other earlier time points (0 hpi up to 24 hpi). Our interpretation is that this over-representation is likely due to the presence of non-infected cells (as your suggestion) at this time point (48 hpi) that were proceeding cyst formation to escape from viral infection. However, this could be also due to the response of infected cells, so that infected cells lock viruses in cyst wall to prevent dissemination new viruses.

As for the cell (healthy & infected) and virus counts, unfortunately, we did not perform such an experiment in our study. By repeating an experiment starting with MOI at 2.88, we may be able to realize a similar infection experiment, take subsamples and count viruses and apparently healthy and infected cells based on the morphology of the cell. However, there would still be a great ambiguity in distinguishing early infected cells from non-infected cells. Furthermore, even with this virus/cell counts, we will still have a difficulty in determining the cause of the host cell transcriptomic dynamics (either the response of healthy cells or the response of infected cells). To our understanding this will require other experiments such as single cell transcriptomics.

Therefore, we consider doing the cell count by repeating a similar experiment may not be sufficient to solve the issue and decided not to perform that.

Following these new considerations inspired by your comments, we have amended the corresponding text to clarify our standpoint and the limitation of this study as follows.

(LL.417-424) “We also found an over-representation of encystment-related genes at 48 hpi (Supplemental Material 2, Fig. S6). As the culture may be a mixture of infected and uninfected amoeba cells at this time point with the initial MOI of 2.88, determining the cause of this over-representation (i.e., due to either healthy or infected cells) requires further investigation. Of note, encystment of both infected and healthy *Veramoeba vermiformis* cells has been observed upon infection by *Faustovirus meriensis* and has been suggested as an antiviral mechanism of the host trapping the viruses inside the cyst walls (64). A similar host strategy may be working for the *A. castellanii*-medusavirus infection system.”

Line 67: medusavirus stheno - 'medusavirus' should be capitalized.

We do not agree with you regarding this. Since we use the name ‘medusavirus stheno’ as a virus name, we prefer using non-capitalized form to comply with the ICTV naming principle below. We are aware that the ICTV rule is sometimes violated in scientific literature, but we consider that complying with the rule is favorable to distinguish taxonomic names from virus names.

<https://talk.ictvonline.org/information/w/faq/386/how-to-write-virus-species-and-other-taxa-names>

Line 117: What was the reason to use the MOI 2.88 and not an MOI of 1 (or other values)? Please specify the rationale. Thanks!

Infection of medusavirus was associated with the appearance of the host amoebas forming cysts at MOI about 1 to 2 in a previous study (Yoshikawa et al., 2019; <https://doi.org/10.1128/JVI.02130-18>). With the aim of characterizing this phenomenon, we performed the infection with a similar MOI level. We added this explanation in the revised version of our manuscript.

(LL.117-119) “In a previous study, infection of medusavirus was associated with the appearance of the host amoebas forming cysts at MOI about 1 to 2 (2). With the aim of investigating this phenomenon, we performed our infection experiment with a similar MOI level.”

Line 157: Why did the authors choose the 'second-best' prediction removing the plastid prediction? The way I understand, the prediction with highest probability should be chosen. I understand that for amoeba infecting viruses, 'plastid' prediction likely is a false prediction. In that case the best approach should be to avoid/remove these genes from further analysis where plastid signal was predicted -rather than choosing the second best hit. At the very least, the authors should explain this caveat in their methods - and also provide the number/percent of such cases for each genome where they found plastid to be the best match and chose the second best

hits

instead.

Thank you for your comment. First, genes predicted to be localized in the plastid only correspond to a small proportion of genes (12/461 (2.6%) of medusavirus genes). Therefore, we believe that the inclusion of these predictions or not (by omitting them or by taking the second predictions) would not affect our main observations such as many medusavirus genes targeting the nucleus.

That being said and considering your comment, we acknowledge that how the cases of plastid prediction should be treated is not straightforward. In the revised version of our manuscript, we have changed to use the results as they are including the “plastid” predictions, because even though these viruses were isolated with amoeba co-culture, there remains a possibility that their natural hosts possess plastid (i.e., photosynthetic organisms, as many other isolated NCLDV). This standpoint is now explained in the Materials and Methods section.

(LL.164-167) “A minor proportion of genes (0.0-5.0% for each virus) were predicted to target the plastid. Although amoebas do not possess plastids, we kept these predictions as they are, because even though these viruses were isolated using amoeba co-culture, there remains a possibility that their natural hosts possess plastids.”

Fig S1: the authors included viruses that not only infect amoeba but other hosts in this analysis (for example tetraselmis virus-1). They should correct this in the manuscript, tables and figures - wherever it is necessary.

Thank you for pointing out this. We have removed viruses that infect non-amoebal microorganisms from Fig. S1 and Data Set 7 for the sake of simplicity.

Line 190-194: The classification of genes in early, late or intermediate needs to be more specific. For example, the authors say that "cluster 2 ("early"), genes that start to be expressed at 1-2 hpi" This will mean that these genes showed no expression before that time point, and there was no RNA-seq reads mapped on these genes before the specified time point. Is that true? If these genes showed some expression before 1-2 hr, then this statement will be incorrect. Based on the data, the authors might want to use a more general language - for example, early genes showed gradual increase in expression starting at 1 hr.

The statements in these paragraphs should be modified as such - or if the authors have alternative explanations, that should be mentioned.

The text has been modified following your suggestions (i.e., “gradual increase”). The corresponding paragraph in the revised version of our manuscript reads as follows.

(LL.194-202) “All viral genes were gradually expressed and continuously increased up to 16 hpi (Fig. 2A). We identified five clusters of viral gene expression profiles using the k-means method (Fig. 2B) and named these clusters as follows: cluster 1 (“immediate early”) genes showed gradual increase in expression from 0 hpi; cluster 2 (“early”) genes showed gradual increase in expression from 1 hpi; clusters 3 and 4 (“intermediate”) genes showed gradual increase in expression from 2 hpi; and cluster 5 (“late”) genes showed gradual increase in expression from 4

hpi. The expression patterns of genes in clusters 3 and 4 were only slightly different; genes in cluster 3 showed higher z-score scaled RPKM values at 8 hpi than those in cluster 4. In the following text, both of these clusters were referred to as “intermediate” genes.”

Reviewer #2 (Comments for the Author):

In their manuscript entitled RNA-seq of medusavirus suggests remodeling of the host nuclear environment at an early infection stage", Zhang et al. performed a time course RNA-seq analysis of *Acanthamoeba castellanii* cells infected by medusavirus. The manuscript describes the results in a clear and concise manner and is well-organized with figures highlighting their main findings. In my opinion, the manuscript would benefit in general having some conclusions for each result paragraph and maybe, statements from the authors or hypothesis to be tested in the future. I must state that I am not qualified to assess the technical set-up of their RNA-seq analysis and have some minor comments regarding the manuscript:

1. In the first paragraph of the introduction, there is a confusion between giant viruses and NCLDV's that also include large DNA viruses like poxviruses, asfarviruses, etc. It would help if the authors would refer either to NCLDV's and then giant viruses in particular.

We introduced a few modifications in the text in order to avoid mixing up the notion of “giant viruses” and the classification NCLDV's (Nucleocytoviricota). The relevant part of the new text reads as follows.

(LL.51-53) “Giant viruses are characterized by their large viral particles and complex genomes, and are found worldwide (1–6). They have been classified within the phylum Nucleocytoviricota (also referred to as Nucleo-Cytoplasmic Large DNA Viruses, NCLDV's) (7).”

2. Line 63: "cosmopolitan" should be "ubiquitous"

Corrected.

3. In the second paragraph of the introduction, it is unclear to me if the authors refer to replication as the viral replication cycle including genome replication and particle assembly or only to replication of the viral genome. Please, specify.

In most of this part, we used the word “replication” to refer to the whole infection cycle from genome replication to particle assembly in the previous version of our manuscript. To make this point clearer, we have modified the text. We now refer to viral genomic DNA replication as “DNA replication” and the whole infection cycle as “viral replication”.

(LL.68-72) “During the infection cycle of medusavirus, its genome enters the host nucleus to initiate DNA replication, and particle assembly and DNA packaging are carried out in the cytoplasm. Of note, the host nuclear membrane remains intact until near the end of viral replication cycle, which represents a unique feature of medusavirus among currently characterized amoeba-infecting giant viruses.”

4. In the materials and methods, the methods for production, purification and titration of the virus should be explained.

Methods for production and purification of viruses have been described in a previous study (Yoshikawa et al., 2019: <https://doi.org/10.1128/JVI.02130-18>). We have added the related reference to the corresponding part. Virus titer was measured by TCID50 by inoculating fresh amoeba on a 96-well plate with serially diluted virus solution. The corresponding text has been modified as follows.

(LL.113-117) “The *A. castellanii* cells were cultured in eight 75-cm² flasks with 25 mL of peptone-yeast-glucose (PYG) medium at 26°C for 1 hour, then infected with purified medusavirus as previously described (2), at a multiplicity of infection (MOI) of 2.88. Titer of medusavirus was measured by TCID50 by inoculating fresh amoeba solution on a 96-well plate with a serially diluted virus solution (29).”

5. Lines 146-148: it is unclear to me if the viral DNA is first observed at 14 hpi or only observed in general. Regarding the new virions, are they also starting to be released at 14 hpi? And how are the viral DNA and new virion detections performed?

These descriptions refer to the previous observations (Yoshikawa et al., 2019; <https://doi.org/10.1128/JVI.02130-18>). These timings are based on the observation of viral DNA in infected cells using fluorescence in situ hybridization analysis. These correspond to the “first” observation but these timings are approximate and not very rigorous ones. Medusavirus was reported to replicate its genome inside the host nucleus, viral DNA can be observed in the cytoplasm at 14 hpi and new virions were released from the host cells at 14 hpi. Above all indicates that the infection cycle finishes at around 14 hpi. We modified the text to clarify these points.

(LL.147-151) “We chose the library from 0–16 hpi to cluster viral genes, because a previous study indicated that replicated viral DNA was first observed in the cytoplasm at approximately 14 hpi and new virions were also observed to be released at the same time point (2), which indicated the termination of a cycle of infection at this time point.”

6. Lines 153-154: could the author give some information on the mitochondrial genes?

We also used RPKM to measure the expression level of mitochondrial genes. We added a description about this treatment.

(LL.157-159) “We did not perform clustering of the expression of mitochondrial genes but analyzed expression of individual genes based on RPKM values.”

7. Lines 191-194: the authors need to comment a bit more on the difference between clusters 3 and 4, the current sentence is quite vague and the separation between these two clusters, which have the same expression profiles, need to be better justified.

Clusters were objectively determined by k-means method only based on the expression pattern of viral genes. Their expression patterns are indeed slightly different (higher z-scored RPKM values at 8 hpi for cluster 3 than for cluster 4, as described in the text). However, we acknowledge that the difference in their expression patterns is not prominent. We did not detect any specific sequence motif to either of them. Though cluster 4 genes are less functionally annotated than cluster 3, they both contain genes belonging to various functional categories. Thus, we decided to call both of them as "intermediate" genes. We added a sentence detailing this situation.

(LL.199-202) “The expression patterns of genes in clusters 3 and 4 were only slightly different; genes in cluster 3 showed higher z-score scaled RPKM values at 8 hpi than those in cluster 4. In the following text, both of these clusters were referred to as “intermediate” genes.”

8. In Fig. 3A, the legend mentions genes with "known" and "unknown" functions, while in the figure panel, it is referred to as "categorized" and "others/unknown". What is the difference between these categories? It should be consistent.

Thank you for pointing this out. We have corrected the text in the figure legend accordingly.

9. In Fig. 3B, a reference is missing regarding the analysis of proteins found in virions.

Thank you. We have added the corresponding reference.

10. Line 207: the authors should be more specific about the bacteriophage (which viral family, etc.).

Thank you for your comment. We have removed this reference as well as the corresponding text from the revised version of our manuscript. At this part of text, we intended to state that the cluster 3 consists of genes from various functional categories, including a viral Yqaj recombinase. However, we found that the paper cited (Vellani & Myers, 2003, <https://doi.org/10.1128/JB.185.8.2465-2474.2003>) in the previous manuscript does not relate to this point. Vellani & Myers (2003) reported the functional characterization of a lambda-exonuclease like exonuclease in *Bacillus subtilis* phage SPP1 and proposed that the exonuclease-synaptase system is essential for the homologous recombination of dsDNA viruses. Therefore, the “essentiality” of Yqaj was not demonstrated in this paper. Thus, we simplified and modified our text as below.

(LL.211-214) “Cluster 3 contained genes in various functional categories, including histone genes (the four core histone genes H2A, H2B, H3, and H4); “DNA replication, recombination, and repair” category (e.g., two of five nuclease genes, a Yqaj viral recombinase gene and a Holliday junction resolvase gene); ...”

11. Lines 219-239: the word "target" should be replaced by "localized" or "specific" for each cellular compartment.

Texts have been amended accordingly (see LL.225-244).

12. Line 235: what does "vacuole" refer to?

We followed the classification of subcellular localizations defined in DeepLoc. In their manuscript (Almagro Armenteros JJ, et al., 2017; <https://doi.org/10.1093/bioinformatics/btx431>) "Lysosome/Vacuole" is explained to be "Contractile, lytic and protein storage vacuole, vacuole lumen and membrane, lysosome lumen and membrane".

13. In Fig. 4, label the Y axis more specifically, e.g. "Proportion of genes".

Corrected

14. In Fig. 7, label the Y axis more specifically, e.g. "Proportion of genes with the different upstream motifs".

Corrected

15. Line 264: should refer to figure 7 not 6, I believe.

Thank you. Corrected.

16. Line 365: a reference is missing.

Yes, the corresponding part of the text has been modified.

(LL.364-366) "We found that linker histone H1, which is not packaged in viral particles (2), was transcribed immediately after the beginning of transcription. In contrast, the four core histones, which are carried in virions (2), started to be transcribed later."

17. Line 394: "recognizing" should be "recognized"

Corrected

18. In general, the idea that the temporal clusters correspond to a classification based on the start of gene expression which is then maintained and continuously increasing up to 16 hpi should be introduced.

Thank you for your advice. We modified the text accordingly

(LL.194-199) "All viral genes were gradually expressed and continuously increased up to 16 hpi (Fig. 2A). We identified five clusters of viral gene expression profiles using the k-means method (Fig. 2B) and named these clusters as follows: cluster 1 ("immediate early") genes showed gradual increase in expression from 0 hpi; cluster 2 ("early") genes showed gradual increase in expression from 1 hpi; clusters 3 and 4 ("intermediate") genes showed gradual increase in expression from 2 hpi; and cluster 5 ("late") genes showed gradual increase in expression from 4 hpi."

19. Line 452: after "suggesting" a "that" is missing.

Corrected

Reviewer #3 (Comments for the Author):

The manuscript by Zhang R et al. is an good paper about RNA-seq of the fascinating medusavirus that has, contrary to other giant viruses, the capability to enter host nucleus. I have no specific modifications to suggests but believe that authors could improve slightly the discussion by presenting Marseilleviruses, the family of viruses harboring histone genes and for which two recent papers have revealed the structure (doi: <https://doi.org/10.1101/2021.04.29.441998>) (<https://doi.org/10.1038/s41594-021-00585-7>)

Thank you. We have modified discussion about viral-encoded histone proteins. Please see the second paragraph of discussion.

(LL.366-374) “The different transcriptional profiles between the linker histone H1 and core histones suggest different functional roles between them. Histone H1 may cooperate with high-mobility group proteins in viral particles to regulate the accessibility of the viral genome for the subsequent transcription process (51, 52), or it may function to regulate the host chromatin. Regarding viral core histone proteins, the core histone proteins of marseilleviruses have been shown to bind DNA and form a structure resembling eukaryotic nucleosomes (53, 54). Marseillevirus histones have been also shown to localize the cytoplasmic viral factories and mature virions in the end of infection (54). Medusavirus core histones may function in a similar way for viral genome packaging as in marseilleviruses.”

August 16, 2021

Prof. Hiroyuki Ogata
Kyoto University
Institute for Chemical Research
Uji, Kyoto 611-0011
Japan

Re: Spectrum00064-21R1 (RNA-seq of medusavirus suggests remodeling of the host nuclear environment at an early infection stage)

Dear Prof. Hiroyuki Ogata:

Congratulations, your revised manuscript has been accepted, and I am forwarding it to the ASM Journals Department for publication. You will be notified when your proofs are ready to be viewed.

Sincerely,

Samuel Campos
Editor, Microbiology Spectrum

Journals Department
DATASET S1, DATASET S2, DATASET S3, DATASET S4, DATASET S5, DATASET S6, DATASET

S7, DATASET S8: Accept

Fig. S1, Fig. S2, Fig. S3, Fig. S4, Fig. S5, Fig. S6: Accept

Table S1, Table S2, Table S3: Accept